# HYPERPARAMETER TUNING AND
# IMPLICIT REGULARIZATION IN MINIBATCH SGD

## ABSTRACT

This paper makes two contributions towards understanding how the hyperparameters of stochastic gradient descent affect the final training loss and test accuracy of neural networks. First, we argue that stochastic gradient descent exhibits two regimes with different behaviours; a *noise dominated* regime which typically arises for small or moderate batch sizes, and a *curvature dominated* regime which typically arises when the batch size is large. In the noise dominated regime, the optimal learning rate increases as the batch size rises, and the training loss and test accuracy are independent of batch size under a constant epoch budget. In the curvature dominated regime, the optimal learning rate is independent of batch size, and the training loss and test accuracy degrade as the batch size rises. We support these claims with experiments on a range of architectures including ResNets, LSTMs and autoencoders. We always perform a grid search over learning rates at all batch sizes. Second, we demonstrate that small or moderately large batch sizes continue to outperform very large batches on the test set, even when both models are trained for the same number of steps and reach similar training losses. Furthermore, when training Wide-ResNets on CIFAR-10 with a constant batch size of 64, the optimal learning rate to maximize the test accuracy only decays by a factor of 2 when the epoch budget is increased by a factor of 128, while the optimal learning rate to minimize the training loss decays by a factor of 16. These results confirm that the noise in stochastic gradients can introduce beneficial implicit regularization.

## 1 INTRODUCTION

Stochastic gradient descent (SGD) is the most popular optimization algorithm in deep learning, but it remains poorly understood. A number of papers propose simple scaling rules that predict how changing the learning rate and batch size will influence the final performance of popular network architectures (Hoffer et al., 2017; Goyal et al., 2017; Smith et al., 2017; Jastrzębski et al., 2017). Some of these scaling rules are contradictory, and Shallue et al. (2018) argue that none of these simple prescriptions work reliably across multiple architectures. Some papers claim SGD with Momentum significantly outperforms SGD without Momentum (Sutskever et al., 2013), but others observe little difference between both algorithms in practice (Kidambi et al., 2018; Zhang et al., 2019).

We hope to clarify this debate. We argue that minibatch stochastic gradient descent exhibits two regimes with different behaviours: a *noise dominated* regime and a *curvature dominated* regime (Ma et al., 2017b; McCandlish et al., 2018; Liu & Belkin, 2018). The noise dominated regime typically arises for small or moderate batch sizes, while the curvature dominated regime typically arises when the batch size is large. The curvature dominated regime may also arise if the epoch budget is small or the loss is poorly conditioned (McCandlish et al., 2018). Our extensive experiments demonstrate that,

1. In the noise dominated regime, the final training loss and test accuracy are independent of batch size under a constant epoch budget, and the optimal learning rate increases as the batch size rises. In the curvature dominated regime, the optimal learning rate is independent of batch size, and the training loss and test accuracy degrade with increasing batch size. The critical learning rate which separates the two regimes varies between architectures.

2. If specific assumptions are satisfied, then the optimal learning rate is proportional to batch size in the noise dominated regime. These assumptions hold for most tasks. However we

observe a square root scaling rule when performing language modelling with an LSTM. This is not surprising, since consecutive gradients in a language model are not independent.

3. SGD with Momentum and learning rate warmup do not outperform vanilla SGD in the noise dominated regime, but they can outperform vanilla SGD in the curvature dominated regime.

There is also an active debate regarding the role of stochastic gradients in promoting generalization. It has been suspected for a long time that stochastic gradients sometimes generalize better than full batch gradient descent (Heskes & Kappen, 1993; LeCun et al., 2012). This topic was revived by Keskar et al. (2016), who showed that the test accuracy often falls if one holds the learning rate constant and increases the batch size, even if one continues training until the training loss ceases to fall. Many authors have studied this effect (Jastrzębski et al., 2017; Smith & Le, 2017; Chaudhari & Soatto, 2018), but to our knowledge no paper has demonstrated a clear generalization gap between small and large batch training under a constant step budget on a challenging benchmark while simultaneously tuning the learning rate. This phenomenon has also been questioned by a number of authors. Shallue et al. (2018) argued that one can reduce the generalization gap between small and large batch sizes if one introduces additional regularization (we note that this is consistent with the claim that stochastic gradients can enhance generalization). Zhang et al. (2019) suggested that a noisy quadratic model is sufficient to describe the performance of neural networks on both the training set and the test set.

In this work, we verify that small or moderately large batch sizes substantially outperform very large batches on the test set in some cases, even when compared under a constant step budget. However the batch size at which the test accuracy begins to degrade can be larger than previously thought. We find that the test accuracy of a 16-4 Wide-ResNet (Zagoruyko & Komodakis, 2016) trained on CIFAR-10 for 9725 updates falls from 94.7% at a batch size of 4096 to 92.8% at a batch size of 16384. When performing language modelling with an LSTM on the Penn TreeBank dataset for 16560 updates (Zaremba et al., 2014), the test perplexity rises from 81.7 to 92.2 when the batch size rises from 64 to 256. We observe no degradation in the final training loss as the batch size rises in either model.

These surprising results motivated us to study how the optimal learning rate depends on the epoch budget for a fixed batch size. As expected, the optimal test accuracy is maximized for a finite epoch budget, consistent with the well known phenomenon of early stopping (Prechelt, 1998). Meanwhile the training loss falls monotonically as the epoch budget increases, consistent with classical optimization theory. More surprisingly, the learning rate that maximizes the final test accuracy decays very slowly as the epoch budget increases, while the learning rate that minimizes the training loss decays rapidly.[1] These results provide further evidence that the noise in stochastic gradients can enhance generalization in some cases, and they suggest novel hyper-parameter tuning strategies that may reduce the cost of identifying the optimal learning rate and optimal epoch budget.

We describe the noise dominated and curvature dominated regimes of SGD with and without Momentum in section 2. We focus on the analogy between SGD and stochastic differential equations (Gardiner et al., 1985; Welling & Teh, 2011; Mandt et al., 2017; Li et al., 2017), but our primary contributions are empirical and many of our conclusions can be derived from different assumptions (Ma et al., 2017b; Zhang et al., 2019). In section 3, we provide an empirical study of the relationship between the optimal learning rate and the batch size under a constant epoch budget, which verifies the existence of the two regimes in practice. In section 4, we study the relationship between the optimal learning rate and the batch size under a constant step budget, which confirms that stochastic gradients can introduce implicit regularization enhancing the test set accuracy. Finally in section 5, we fix the batch size and consider the relationship between the optimal learning rate and the epoch budget.

## 2 THE TWO REGIMES OF SGD

Full batch gradient descent is in the "curvature dominated" regime, where the optimal learning rate is determined by the curvature of the loss function. The full batch gradient descent update on the $i^{th}$ step is given by $\omega_{i+1} = \omega_i - \epsilon \frac{dC}{d\omega}\big|_{\omega=\omega_i}$, where the loss $C(\omega) = \frac{1}{N}\sum_{j=1}^{N} C(\omega, x_j)$ is a function of the parameters $\omega$ and the training inputs $\{x_j\}_{j=1}^{N}$, and $\epsilon$ denotes the learning rate. In order to minimize the loss as quickly as possible, we will set the learning rate at the start of training as large as

---

[1]These conclusions might change if the number of updates is very small, in which case the optimal learning rate schedules on the training test and the test set may agree.

we can while avoiding divergences or instabilities. To build our intuition for this, we approximate the loss by a strictly convex quadratic, $C(\omega) \approx \frac{1}{2}\omega^T H\omega$. For simplicity, we assume the minimum lies at $\omega = 0$. Substituting this approximation into the parameter update, we conclude $\omega_{i+1} = \omega_i - \epsilon \omega_i^T H$. In the eigenbasis of $H$, $\omega_{i+1} = \omega_i(I - \Lambda)$, where $I$ denotes the identity matrix and $\Lambda$ denotes a diagonal matrix comprising the eigenvalues of $H$. The iterates will converge so long as the learning rate $\epsilon < \epsilon_{crit}$, where $\epsilon_{crit} = 2/\lambda_{max}$ is the critical learning rate above which training diverges, and $\lambda_{max}$ is the largest Hessian eigenvalue. We call this inequality the *curvature constraint*, and the optimal initial learning rate with full batch gradients will be just below $\epsilon_{crit}$. For clarity, although the critical learning rate will perform poorly for high curvature directions of the loss, we can introduce learning rate decay to minimize the loss along these directions later in training (Ge et al., 2019). Of course, in realistic loss landscapes this critical learning rate might also change during training.

Acceleration methods such as Heavy-Ball Momentum (referred to as Momentum from here on) (Polyak, 1964) were designed to enable faster convergence on poorly conditioned loss landscapes. Momentum works by taking an exponential moving average of previous gradients, $\omega_{i+1} = \omega_i - \epsilon \sum_{j=0}^{i} m^{i-j} \frac{dC}{d\omega}\big|_{w=w_i}$, where $m$ denotes the momentum coefficient. Gradients in high curvature directions, which often switch sign between updates, partially cancel out. This enables Momentum to take larger steps in low curvature directions while remaining stable in high curvature directions. On quadratic losses for example, Momentum increases the critical learning rate: $\epsilon_{crit} \leq 2(1+m)/\lambda_{max}$ (Goh, 2017), and can minimize the training loss in fewer steps than full batch gradient descent.

In practice we do not compute a full batch gradient, we estimate the gradient over a minibatch (Robbins & Monro, 1951; Bottou, 2010). This introduces noise into our parameter updates, and this noise will play a crucial role in the training dynamics in some cases. However when the batch size is large, and the number of training epochs is finite, the noise in the parameter updates is low, and so typically most of training is governed by the curvature of the loss landscape (similar to full batch gradient descent). We call this large batch training regime *curvature dominated*. When the batch size is in the curvature dominated regime, we expect the optimal initial learning rate to be determined by the critical learning rate $\epsilon_{crit}$, and for SGD with Momentum to outperform SGD without Momentum. On the other hand, when the batch size is small, typically most of the training process is governed by the noise in the parameter updates, and we call this small batch training regime *noise dominated*.

In order to build a model of the training dynamics in the noise dominated regime, we must make some assumptions. Following previous work (Mandt et al., 2017; Li et al., 2017; Smith & Le, 2017; Jastrzębski et al., 2017; Simsekli et al., 2019), we assume the gradients of individual examples are independent samples from an underlying distribution, and that this distribution is not heavy tailed. When the training set size $N \gg B$, the batch size $B \gg 1$, and $B \ll N$, we can apply the central limit theorem to model the noise in a gradient update by a Gaussian noise source $\delta$, whose covariance is inversely proportional to the batch size. We therefore approximate the SGD update by,

$$(\omega_{i+1} - \omega_i) \approx -\epsilon\left(\frac{dC}{d\omega}\Big|_{\omega=\omega_i} + \frac{\delta_i}{\sqrt{B}}\right), \tag{1}$$

To interpret this update, we introduce the temperature $T = \epsilon/B$ to obtain,

$$(\omega_{i+1} - \omega_i) \approx -\epsilon\frac{dC}{d\omega}\Big|_{\omega=\omega_i} + \sqrt{\epsilon T}\delta_i. \tag{2}$$

Equation 2 describes the discretization of a stochastic differential equation with step size $\epsilon$ and temperature $T$ (Gardiner et al., 1985), and we expect the dynamics of SGD to follow this underlying stochastic differential equation, so long as the learning rate $\epsilon \ll \epsilon_{crit}$ and the assumptions above are satisfied. When equation 2 holds and the learning rate $\epsilon \ll \epsilon_{crit}$, any two training runs with the same temperature and the same epoch budget should achieve similar performance on both the training set and the test set. Consequently, we usually expect the learning rate to scale linearly with the batch size in the noise dominated regime, and this was observed in many empirical studies (Krizhevsky, 2014; Goyal et al., 2017; Smith et al., 2017; Jastrzębski et al., 2017; McCandlish et al., 2018). For completeness, we derive this linear scaling rule in appendix B, and we demonstrate that the linear scaling rule can be derived without assuming that the batch size $B \gg 1$. However the remaining assumptions above are required, and they are not always satisfied. This linear scaling rule is therefore not valid in all cases (Shallue et al., 2018; Simsekli et al., 2019). Empirically however, we have found that all batch sizes in the noise dominated regime achieve similar test accuracies and training losses under a constant epoch budget, even when the optimal learning rate does not obey linear scaling.

**Further observations on the two regimes:** Many previous works have established that SGD with and without Momentum are equivalent in the small learning rate limit when $m$ is fixed (Orr & Leen, 1994; Qian, 1999; Yuan et al., 2016). In this limit, the speed of convergence of SGD with Momentum is governed by the effective learning rate $\epsilon_{eff} = \epsilon/(1-m)$, and the temperature $T = \epsilon_{eff}/B$ (Mandt et al., 2017; Smith & Le, 2017). We therefore expect SGD with and without Momentum to achieve the same final training losses and test accuracies in the noise dominated regime (where the optimal learning rate is smaller than $\epsilon_{crit}$). Supporting this claim, Shallue et al. (2018) confirmed in a recent empirical study that SGD with and without Momentum achieve similar test accuracies in the small batch limit, while SGD with Momentum outperforms vanilla SGD in the large batch limit.

In recent years, a number of authors have exploited large batch training and parallel computation to minimize the wallclock time of training deep networks (De et al., 2016; Smith et al., 2017; Akiba et al., 2017; You et al., 2018; McCandlish et al., 2018; Shallue et al., 2018). Goyal et al. (2017) succeeded in training ResNet-50 to over 76% accuracy in under one hour, and since then this has fallen to just a few minutes (Mikami et al., 2018; Ying et al., 2018). Goyal et al. (2017) also introduced learning rate warmup, and found that it enabled stable training with larger batch sizes; a result which we confirm in this work. This procedure has a straightforward interpretation within the two regimes. If the critical learning rate increases early in training, then learning rate warmup will increase the largest stable learning rate, which in turn enables efficient training with larger minibatches.

**Learning rate schedules:** In the noise dominated regime, the learning rate increases as the batch size rises, and therefore as we increase the batch size we will eventually invalidate the assumption $\epsilon \ll \epsilon_{crit}$ and enter the curvature dominated regime. There may be a transition phase between the two regimes (Liu & Belkin, 2018), although our experiments suggest this transition can be surprisingly sharp in practice. We note that, with the optimal learning rate schedule, many batch sizes might exhibit both the curvature dominated regime (typically early in training) and the noise dominated regime (typically towards the end of training). For example, on simple quadratic loss landscapes at any batch size, the optimal learning rate schedule for minimizing the *training loss* begins with an initial learning rate close to $\epsilon_{crit}$, followed by learning rate decay (Zhang et al., 2019). However in practice it is not possible to identify the optimal learning rate schedule within a realistic computation budget. Practitioners prefer simple learning rate schedules, often parameterized by an initial learning rate and a few sharp drops (He et al., 2016). These schedules are easy to tune and are also thought to generalize well. For these popular schedules, the optimal initial learning rate would be determined by whether most of the training process is noise dominated or curvature dominated. Furthermore, Smith & Le (2017) suggested that there may be an optimal temperature at early times that promotes generalization. If this speculation is correct, then the optimal learning rate schedule to maximize the *test accuracy* would select the noise dominated regime throughout the whole of training when the batch size is small. Of course, if the loss is extremely poorly conditioned, the critical learning rate may already be optimal when the batch size is 1, although we have never seen this in practice[2].

**Implicit regularization:** In the noise dominated regime, the temperature $T$ defines the influence of gradient noise on the dynamics. In the noisy quadratic model, our goal during training is to minimize the effect this noise has on the final parameters (Polyak & Juditsky, 1992; Zhang et al., 2019). However recently many authors have argued that minibatch noise can be beneficial, helping us to select final parameters which perform well on the test set (Keskar et al., 2016; Jastrzębski et al., 2017; Chaudhari & Soatto, 2018). In this alternative perspective, there may be an "optimal temperature" early in training, which drives the parameters towards regions that generalize well (Smith et al., 2017; Park et al., 2019). The noisy quadratic model predicts that increasing the batch size may increase the final training loss under a constant epoch budget, but that it should not increase the final training loss under a constant step budget. Note that this model does not make explicit predictions about the test accuracy. We note that Jin et al. (2019) argued noise enables SGD to escape saddle points in non-convex landscapes, which could enable small batch sizes to achieve lower training losses under both constant epoch and constant step budgets. The optimal temperature perspective predicts that increasing the batch size may increase the final training loss under a constant epoch budget, but it does not predict what will happen to the training loss under a constant step budget. Crucially, it predicts that beyond some threshold batch size (which may be very large), increasing the batch size will decrease the test accuracy under both constant epoch and constant step budgets[3].

---

[2]We assume the epoch budget is large enough to fit the training data. Training may occur in the curvature dominated regime at all batch sizes when the epoch budget is very small (McCandlish et al., 2018).

[3]We emphasize that both perspectives agree that learning rate decay should be used at the end of training.

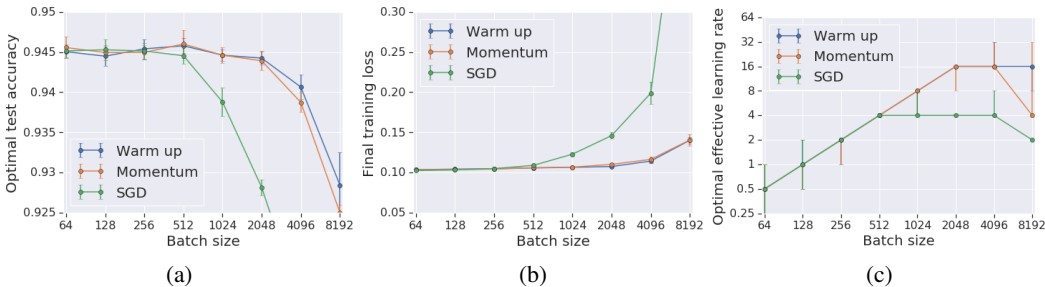

Figure 1: A 16-4 Wide-ResNet, trained with ghost batch normalization on CIFAR-10 for 200 epochs. We report the performance of SGD with and without Momentum, and the combination of SGD with Momentum and learning rate warmup. We perform a grid search to identify the optimal learning rate which maximizes the test accuracy, and report the mean performance of the best 12 of 15 runs. a) The test accuracy is independent of batch size when the batch size is small, but begins to fall when the batch size exceeds 512. b) Similarly, the training loss at the optimal effective learning rate is independent of batch size when the batch size is small, but rises rapidly if acceleration techniques are not used when the batch size is large. c) The optimal effective learning rate is proportional to batch size when the batch size is small, while it is independent of batch size when the batch size is large.

## 3   TRAINING UNDER A CONSTANT EPOCH BUDGET

In order to verify the existence of the two regimes of SGD, we study how the performance on the training and test sets, and the optimal learning rate, depend on the batch size under a constant epoch budget (when using a realistic learning rate decay schedule). To explore this, we must first select a set of model architectures and datasets, and then identify a single learning rate decay schedule that performs well across all of these tasks, matching the baseline performance of the schedules reported in the original papers. For clarity, in the main text we only report experiments using Wide-ResNets on CIFAR-10 (Zagoruyko & Komodakis, 2016), however we provide additional experiments using ResNet-50, LSTMs and autoencoders in appendix D (Sutskever et al., 2013; Zaremba et al., 2014; He et al., 2016). For each experiment in this section, we train for the same number of epochs $N_{epochs}$ reported in the original papers (e.g., 200 epochs on CIFAR-10). Our chosen schedule is the following. We hold the learning rate constant for the first $N_{epochs}/2$ epochs. Then for the remainder of training, we reduce the learning rate by a factor of 2 every $N_{epochs}/20$ epochs. This scheme has a single hyper-parameter, the initial learning rate $\epsilon$, and we found this schedule to reliably meet the performance reported by the authors of the original papers. In some of our experiments we also introduce learning rate warmup (Goyal et al., 2017), whereby the learning rate is linearly increased from 0 to $\epsilon$ over the first 5 epochs of training. We illustrate these schedules in appendix A.

We will evaluate the optimal test accuracy and the optimal learning rate for a range of batch sizes. At each batch size, we train the model 15 times for a range of learning rates on a logarithmic grid. For each learning rate in this grid, we take the best 12 runs and evaluate the mean and standard deviation of their test accuracy. The optimal test accuracy is defined by the maximum value of this mean, and the corresponding learning rate is the optimal learning rate. This procedure ensures our results are not corrupted by outliers or failed training runs. To define error bars on the optimal learning rate, we include any learning rate whose mean accuracy was within one standard deviation of the mean accuracy of the optimal learning rate, and we always verify that both the optimal learning rate and the error bars are not at the boundary of our learning rate grid. We apply data augmentation including padding, random crops and left-right flips. The momentum coefficient $m = 0.9$, the L2 regularization coefficient is $5 \times 10^{-4}$, and we use ghost batch normalization with a ghost batch size of 64 (Hoffer et al., 2017). We also report the mean final training loss at the optimal learning rate. We note that although we tune the learning rate on the test set, our goal in this paper is not to report state of the art performance. Our goal is to compare the performance at different batch sizes and with different training procedures. We apply the same experimental protocol in each case (Shallue et al., 2018). We also provide the full results of a learning rate sweep at two batch sizes in appendix D.

In figure 1a, we plot the optimal test accuracy for a range of batch sizes with a 16-4 Wide-ResNet, trained with batch normalization using SGD with and without Momentum, and also with learning

rate warmup. All three methods have the same optimal test accuracy when the batch size is small, but both SGD with Momentum and learning rate warmup outperform SGD without Momentum when the batch size is large. The optimal test accuracy is independent of batch size when the batch size is small, but begins to fall when the batch size grows. A very similar trend is observed for the final training loss at the optimal effective learning rate in figure 1b. To understand these results, we plot the optimal effective learning rate against batch size in figure 1c (for SGD, $\epsilon_{eff} = \epsilon$). Looking first at the curve for vanilla SGD, the learning rate is proportional to the batch size below $B \approx 512$, beyond which the optimal learning rate is constant. SGD with Momentum and warmup have the same optimal effective learning rate as SGD in the small batch limit, but their optimal effective learning rates are larger when $B > 512$. All of these results exactly match the theoretical predictions we made in section 2.

The behaviour of SGD is strongly influenced by batch normalization (Bjorck et al., 2018; Santurkar et al., 2018; Sankararaman et al., 2019; Park et al., 2019). We therefore repeat this experiment without normalization in appendix D. To ensure training is stable we introduce a simple modification to the initialization scheme, "ZeroInit", introduced in a parallel submission (Anonymous, 2020) and defined for the reader's convenience in appendix C. This modification enables the training of very deep networks, and it reduces the gap in optimal test accuracy between networks trained with and without batch normalization. We observe remarkably similar trends, although the critical learning rate, beyond which the optimal learning rate of SGD is independent of batch size, is significantly smaller. We also provide similar experiments in appendix D for a range of model architectures. In all cases, we observe a transition from a small batch regime, where the learning rate increases with the batch size and SGD with Momentum does not outperform SGD, to a large batch regime, where the learning rate is independent of the batch size and SGD with Momentum outperforms SGD. Under a constant epoch budget, both the training loss and the optimal test accuracy are independent of batch size in the noise dominated regime, but they begin to fall when one enters the curvature dominated regime. In most cases the optimal learning rate in the noise dominated regime was proportional to batch size, however for an LSTM trained on the Penn TreeBank dataset, the optimal learning rate was proportional to the square root of the batch size. To understand this, we note that consecutive minibatches in a language model are correlated, which violates the linear scaling assumptions discussed in section 2.

## 4  TRAINING UNDER A CONSTANT STEP BUDGET

In the previous section, we studied how the optimal learning rate depends on the batch size under a constant epoch budget. As predicted in section 2, we found that SGD transitions between two regimes with different behaviours in a range of popular architectures. However, these results can be explained from a number of different perspectives, including both the interpretation of small learning rate SGD as the discretization of stochastic differential equation (Li et al., 2017; Mandt et al., 2017), and also a simple noisy quadratic model of the loss landscape (McCandlish et al., 2018; Zhang et al., 2019). Crucially, the results of the previous section do not tell us whether minibatch noise introduces implicit regularization that selects parameters that perform better on the test set, since under a constant epoch budget, when we increase the batch size we reduce the number of training steps.

In order to establish whether minibatch noise enhances the test accuracy, we now evaluate how the optimal test accuracy depends on the batch size under a constant step budget. In this scheme, the number of training epochs is proportional to the batch size, which ensures that large batch sizes are able to minimize the training loss. In table 1, we report the optimal test accuracy of our 16-4 Wide-ResNet on CIFAR-10 at batch sizes ranging from 1024 to 16384. For each batch size, we train for 9765 updates using SGD with Momentum. Note that this corresponds to 200 epochs when the batch size is 1024 (to ensure these experiments did not require an unreasonably large epoch budget, we intentionally selected a batch size just below the boundary of the curvature dominated regime in figure 1). Following our previous schedule, we hold the learning rate constant for 4882 updates, and then decay the learning rate by a factor of 2 every 488 steps. We conclude that the optimal test accuracy initially increases with increasing batch size, but it then begins to fall. The optimal test accuracy at batch size 4096 is $94.7\%$, but the optimal test accuracy at batch size 16384 is just $92.8\%$. We also report the final training loss, which falls continuously with batch size. Notice that this occurs despite the fact that the optimal learning rate is defined by the test set accuracy. We observed similar results when training without batch normalization using zeroInit, which are shown in table 2, and we also provide similar results on the autoencoder and LSTM tasks in appendix E.

Table 1: The optimal test accuracy and final training loss for a range of batch sizes under a constant step budget. For each batch size, we train a 16-4 Wide-ResNet with ghost batch normalization for 9765 updates, and we perform a grid search to identify the optimal learning rate which maximizes the test set accuracy. We provide the average performance of the best 12 out of 15 training runs. The final training loss falls as the batch size increases, but the optimal test accuracy drops significantly for batch sizes greater than 4096. This strongly suggests that minibatch noise can enhance generalization.

| Batch size | Optimal test accuracy (%) | Final training loss | Optimal effective learning rate |
|---|---|---|---|
| 1024 | $94.6 \pm 0.1$ | $0.107 \pm 0.001$ | $2^3$ $(2^3$ to $2^3)$ |
| 4096 | $94.7 \pm 0.1$ | $0.025 \pm 0.000$ | $2^4$ $(2^3$ to $2^4)$ |
| 16384 | $92.5 \pm 0.6$ | $0.019 \pm 0.004$ | $2^5$ $(2^4$ to $2^5)$ |

Table 2: The optimal test accuracy and final training loss for a range of batch sizes under a constant step budget. For each batch size, we train a 16-4 Wide-ResNet without batch normalization using "zeroInit" (defined in appendix C) for 156,250 updates. We perform a grid search to identify the optimal learning rate which maximizes the test accuracy, and we provide the average performance of the best 12 out of 15 training runs. The final test accuracy falls for very large batches.

| Batch size | Optimal test accuracy | Final training loss | Optimal effective learning rate |
|---|---|---|---|
| 64 | $94.4 \pm 0.1$ | $0.192 \pm 0.002$ | $2^{-2}$ $(2^{-2}$ to $2^{-1})$ |
| 256 | $94.4 \pm 0.1$ | $0.071 \pm 0.001$ | $2^{-1}$ $(2^{-1}$ to $2^0)$ |
| 1024 | $93.8 \pm 0.1$ | $0.028 \pm 0.000$ | $2^{-1}$ $(2^{-1}$ to $2^0)$ |

These results demonstrate that stochastic gradient noise can enhance generalization, increasing the test accuracy here by nearly $2\%$. This shows that while the noisy quadratic model may help describe the evolution of the training loss, it does not capture important phenomena observed on the test set in popular networks (McCandlish et al., 2018; Zhang et al., 2019). While many previous authors have observed that stochastic gradient noise enhances generalization (Keskar et al., 2016; Smith & Le, 2017; Jastrzębski et al., 2017; Chaudhari & Soatto, 2018; Park et al., 2019), we believe our experiment is the first to demonstrate this effect when training a well-respected architecture to the expected test accuracy with a properly tuned learning rate schedule. Finally, we emphasize that the implicit regularization introduced by stochastic gradients should be considered complementary to the implicit bias of gradient descent (Ma et al., 2017a; Gunasekar et al., 2018a;b; Soudry et al., 2018; Jacot et al., 2018; Allen-Zhu et al., 2018; Oymak & Soltanolkotabi, 2018; Azizan et al., 2019).

## 5 THE DEPENDENCE OF THE LEARNING RATE ON THE EPOCH BUDGET

We established in section 4 that, in some architectures and datasets, the noise introduced by stochastic gradients does enhance generalization. This motivates the following question; if the batch size is fixed, how does the optimal test accuracy and optimal learning rate depend on the epoch budget? In particular, is the optimal training temperature, defined by the ratio of the learning rate to the batch size, independent of the epoch budget, or does it fall as the number of training epochs increases.

To answer this question, we select a batch size of 64, and we evaluate both the optimal test accuracy and the optimal training loss for a range of epoch budgets using SGD with Momentum. As before, we use our standardized learning rate schedule for each epoch budget, described in appendix A. However, to study the effect of the optimal training temperature, we now *independently measure both the optimal learning rate to maximize the test accuracy, and the optimal learning rate to minimize the training loss*. The optimal test accuracy and optimal training loss are shown in figures 2a and 2b. We train both with and without batch normalization, and we provide the optimal learning rates with batch normalization in figure 2c, and the optimal learning rates without batch normalization in figure 2d.

Considering first figure 2a, the optimal test accuracy initially increases as we increase the epoch budget, however with batch normalization it saturates for epoch budgets beyond 800 epochs, while without batch normalization it falls for epoch budgets beyond 400 epochs. This is similar to the well-known phenomenon of early stopping (Prechelt, 1998; Caruana et al., 2001). As expected, in figure 2b, we find that the optimal training loss falls monotonically as the epoch budget increases.

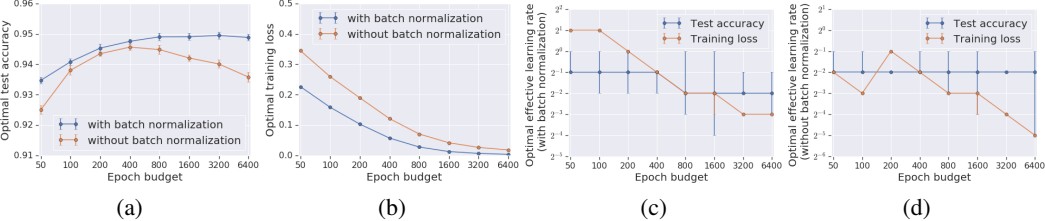

Figure 2: The performance of a 16-4 Wide-ResNet on CIFAR-10 using SGD with Momentum and a batch size of 64. We train both with batch normalization, and also without batch normalization using "zeroInit". We identify both the optimal effective learning rate which maximizes the test accuracy and the optimal effective learning rate which minimizes the training loss, and we present the mean performance of the best 12 out of 15 runs. a) Initially the test accuracy rises as the epoch budget increases, however when training without batch normalization it begins to fall beyond 400 training epochs. b) The training loss falls monotonically as the epoch budget rises. c) With batch normalization, the learning rate which minimizes the training loss falls rapidly as the epoch budget rises, while the learning rate which maximizes the test accuracy only varies by a factor of 2 when the epoch budget rises over two orders of magnitude. d) Similarly, without batch normalization using zeroInit, the learning rate which minimizes the training loss falls as the epoch budget rises while the learning rate which maximizes the test accuracy is constant for all epoch budgets considered.

Figures 2c and 2d are more surprising. Both with and without batch normalization, the learning rate which minimizes the training loss falls rapidly as the epoch budget rises. This is exactly what one would expect from convergence bounds or the noisy quadratic model (Ma et al., 2017b; Zhang et al., 2019). Strikingly however, when training with batch normalization the learning rate which maximizes the test accuracy only falls by a factor of 2 when we increase the epoch budget from 50 to 6400 epochs. Meanwhile when training without batch normalization using zeroInit, the learning rate which maximizes the test accuracy is constant for all epoch budgets considered. These results support the claim that when training deep networks on classification tasks, there is an optimal temperature scale early in training (Smith & Le, 2017; Park et al., 2019), which biases small batch SGD towards parameters which perform well on the test set. Our results also suggest that one might be able to reduce the cost of hyper-parameter tuning by first identifying the optimal learning rate for a modest epoch budget, before progressively increasing the epoch budget until the test accuracy saturates. We provide additional experimental results on the LSTM and autoencoder in appendix F. To further investigate whether there is an optimal temperature early in training, we provide an additional experiment in appendix G where we independently tune both the initial learning rate and the final learning rate in our schedule. Our main claims in this section still hold in this experiment.

## 6 CONCLUSIONS

The contributions of this work are twofold. First, we verified that SGD exhibits two regimes with different behaviours. In the noise dominated regime which arises when the batch size is small, the test accuracy is independent of batch size under a constant epoch budget, the optimal learning rate increases as the batch size rises, and acceleration techniques do not outperform vanilla SGD. Meanwhile in the curvature dominated regime which arises when the batch size is large, the optimal learning rate is independent of batch size, acceleration techniques outperform vanilla SGD, and the test accuracy degrades with batch size. If certain assumptions are satisfied, the optimal learning rate in the noise dominated regime is proportional to batch size. These assumptions hold for most tasks. Second, we confirm that a gap in test accuracy between small or moderately large batch sizes and very large batches persists even when one trains under a constant step budget. When training a 16-4 Wide-ResNet on CIFAR-10 for 9765 updates, the test accuracy drops from 94.7% at a batch size of 4096 to 92.5% at a batch size of 16384. We also find that the optimal learning rate which maximizes the test accuracy of Wide-ResNets depends very weakly on the epoch budget while the learning rate which minimizes the training loss falls rapidly as the epoch budget increases. These results confirm that stochastic gradients introduce implicit regularization which enhances generalization, and they provide novel insights which could be used to reduce the cost of identifying the optimal learning rate.

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

## A   OUR LEARNING RATE DECAY SCHEDULE

For clarity, we illustrate our standardized learning rate decay schedule in figure 3. As specified in the main text, if the epoch budget is $N_{epochs}$, we hold the learning rate constant for $N_{epochs}/2$, before decaying the learning rate by a factor of 2 every $N_{epochs}/20$. When learning rate warmup is included, we linearly increase the learning rate to its maximal value over the first 5 epochs of training.

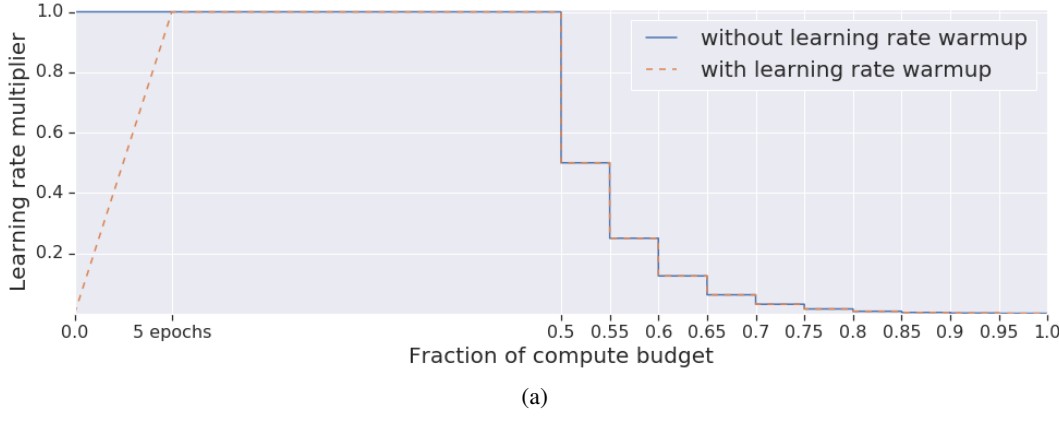

(a)

Figure 3: Our standardized learning rate schedule, both with and without learning rate warmup. Note that while the learning rate decays at points defined by the fraction of the total epoch budget completed, we perform learning rate warmup for 5 epochs, irrespective of the epoch budget.

## B   DERIVING THE LINEAR SCALING RULE FOR SMALL BATCH SIZES

In the main text, we applied the central limit theorem to approximate a single SGD step by,

$$\Delta\omega_i = (\omega_{i+1} - \omega_i) \approx -\epsilon \frac{dC}{d\omega}\Big|_{\omega=\omega_i} + \sqrt{\epsilon T}\delta_i. \tag{3}$$

The temperature $T = \epsilon/B$, $\mathbb{E}(\delta_i) = 0$ and $\mathbb{E}(\delta_i \delta_j^\mathsf{T}) = F(\omega_i)\delta_{ij}$, where $F(\omega)$ is the empirical Fisher information matrix. Equation 5 holds so long as the gradient of each training example is an independent and uncorrelated sample from an underlying short tailed distribution. Additionally, it assumes that the training set size $N \gg 1$, the batch size $B \gg 1$, and $B \ll N$. To derive the linear scaling rule, we consider the total change in the parameters over $n$ consecutive parameter updates,

$$\Delta\omega_i' = \left(\sum_{j=0}^{n-1} \Delta\omega_{i+j}\right) \approx -\epsilon \left(\sum_{j=0}^{n-1} \frac{dC}{d\omega}\Big|_{\omega=\omega_{i+j}}\right) + \sqrt{n\epsilon T}\xi_i \tag{4}$$

The noise $\xi_i = (1/\sqrt{n}) \sum_{j=0}^{n-1} \delta_{i+j}$. When the product of the number of steps $n$ and the learning rate $\epsilon$ is much smaller than the critical learning rate, $n\epsilon \ll \epsilon_{crit}$, the parameters do not move far enough for the gradients to significantly change, and therefore for all $\{j, j'\}$ greater than 0 and less than $n$,

$$\frac{dC}{d\omega}\Big|_{\omega=\omega_{i+j}} \approx \frac{dC}{d\omega}\Big|_{\omega=\omega_i} \tag{5}$$

$$\mathbb{E}(\delta_{i+j}\delta_{i+j'}) \approx F(\omega_i)\delta_{jj'} \tag{6}$$

Equation 5 implies that,

$$\Delta\omega_i' \approx -n\epsilon \frac{dC}{d\omega}\Big|_{\omega=\omega_i} + \sqrt{n\epsilon T}\xi_i. \tag{7}$$

While equation 6 implies that $\mathbb{E}(\xi_i) = 0$ and $\mathbb{E}(\xi_i \xi_i^\mathsf{T}) \approx F(\omega_i)$. We therefore conclude that $\xi$ and $\delta$ are both Gaussian random variables from the same distribution. Comparing equation 3 and equation 7, we conclude that $n$ SGD updates at temperature $T$ with learning rate $\epsilon$ is equivalent to a single SGD step at temperature $T$ with learning rate $n\epsilon$. Since the temperature $T = \epsilon/B$, this implies that

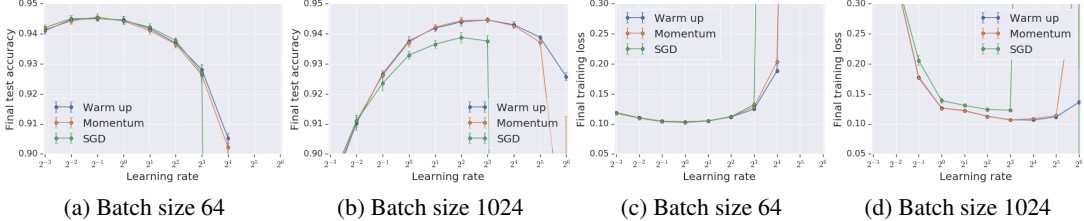

(a) Batch size 64      (b) Batch size 1024      (c) Batch size 64      (d) Batch size 1024

Figure 4: A 16-4 Wide-ResNet, trained with batch normalization on CIFAR-10 for 200 epochs. For completeness, we provide the full results of a learning rate sweep at two batch sizes, 64 and 1024. The smaller batch size is in the noise dominated regime, while the larger batch size is in the curvature dominated regime. We provide the final test accuracy at a range of learning rates in figures a and b, and we provide the final training loss at a range of learning rates in figures c and d. We note that SGD, SGD with Momentum, and SGD with Momentum and learning rate warmup always achieve similar final performance in the small learning rate limit, while SGD performs poorly when the learning rate is large. When the batch size is small, the optimal learning rate is also small, and so all three methods perform similarly. When the batch size is large, the optimal learning rate is also large, and SGD underperforms both SGD with Momentum and SGD with Momentum and learning rate warmup.

when $\epsilon \ll \epsilon_{crit}$, then simultaneously doubling both the learning rate and the batch size should draw samples from the same distribution over parameters after the same number of training epochs.

This prediction is known as the linear scaling rule (Krizhevsky, 2014; Goyal et al., 2017; Mandt et al., 2017; Smith & Le, 2017; Jastrzębski et al., 2017; Chaudhari & Soatto, 2018; McCandlish et al., 2018; Shallue et al., 2018). Since this linear scaling rule assumes that $\epsilon \ll \epsilon_{crit}$, it usually holds when the batch size is small, which appears to contradict the assumption $B \gg 1$ above. Crucially however, the distribution of $\delta_i$ does not matter in practice, since our dynamics is governed by the combined influence of noise over multiple consecutive updates, $\xi_i = (1/\sqrt{n}) \sum_{j=0}^{n-1} \delta_{i+j}$. In other words, we do not require that equation 3 is an accurate model of an single SGD step, we only require that equation 7 is an accurate model of $n$ SGD steps. We therefore conclude that $\delta_i$ does not need to be Gaussian, we only require that $\xi_i$ is Gaussian. The central limit theorem predicts that, if $\delta_i$ is an independent random sample from a short-tailed distribution, $\xi_i$ will be Gaussian if $N \gg 1$, $nB \gg 1$ and $nB \ll N$. If $\epsilon \ll \epsilon_{crit}$, then we can choose $1 \ll n \ll N$, and discard the assumption $B \gg 1$.

## C  ZeroInit, a simple initialization scheme for residual networks

In the main text, we provided a number of experimental results on wide residual networks, trained without batch normalization using "ZeroInit". This simple initialization scheme, introduced in a parallel submission (Anonymous, 2020), is now presented here for the reader's convenience. The scheme comprises three simple modifications to the original Wide-ResNet architecture (Zagoruyko & Komodakis, 2016):

1. It introduces a scalar multiplier at the end of each residual branch, just before the residual branch and skip connection merge. This scalar multiplier is initialized to zero.

2. It introduces biases to each convolutional layer, which are initialized to zero.

3. It introduces dropout on the final fully connected layer.

Modification 1 is sufficient to train very deep residual networks without batch normalization, while modifications 2 and 3 slightly increase the final test accuracy. We emphasize that dropout is included solely to illustrate that additional regularization is required when batch normalization is removed. Dropout itself is not required to train without batch normalization, and similar or superior results could be obtained with alternative regularizers. Throughout this paper, the drop probability is 60%.

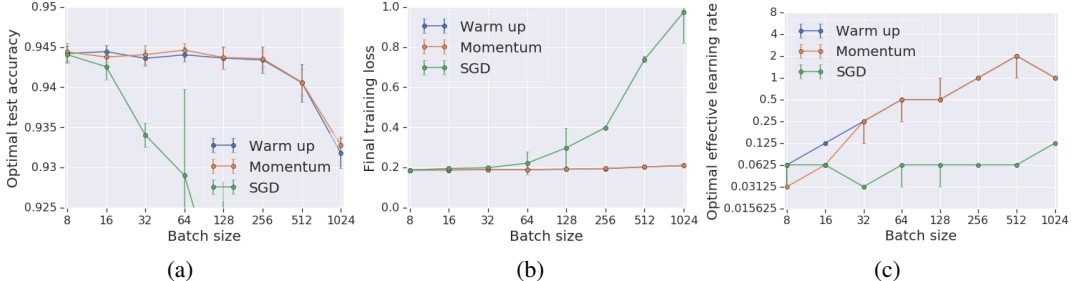

Figure 5: A 16-4 Wide-ResNet, trained without batch normalization using zeroInit on CIFAR-10 for 200 epochs. We report the performance of SGD, SGD w/ Momentum, and SGD w/ momentum using learning rate warmup. We perform a grid search to identify the optimal learning rate which maximizes the test accuracy, and report the mean performance of the best 12 of 15 runs. a) The test accuracy of SGD w/ Momentum is independent of batch size when the batch size is small, but begins to fall when the batch size exceeds 256. The test accuracy of vanilla SGD is falling for all batch sizes considered. b) The training loss at the optimal effective learning rate is independent of batch size when the batch size is small, but rises rapidly if acceleration techniques are not used when the batch size is large. c) The optimal effective learning rate is proportional to batch size when the batch size is small for SGD w/ Momentum, while it is independent of batch size for vanilla SGD.

## D  ADDITIONAL RESULTS UNDER A CONSTANT EPOCH BUDGET

Here we provide some additional results studying how the optimal effective learning rate depends on the batch size under a constant epoch budget.

**Additional results on 16-4 Wide-ResNet on CIFAR with batch normalization.** In figure 4, we provide additional results with the 16-4 Wide-ResNet, trained with batch normalization on CIFAR-10 for 200 epochs. Here we provide the final test set accuracies and the final training set losses for a full learning rate sweep at two batch sizes, 64 and 1024. From figure 4, we see that SGD, SGD with Momentum and warm up learning rates always achieve similar final performance in the small learning rate limit. This confirms previous theoretical work showing the equivalence of SGD and SGD with momentum in the small learning rate limit when the momentum parameter is kept fixed (Orr & Leen, 1994; Qian, 1999; Yuan et al., 2016). Further, we see that SGD performs poorly compared to SGD with Momentum and warmup learning rates when the learning rate is large. When the batch size is small, the optimal learning rates for all three methods are also small, and so all three methods perform similarly. On the other hand, when the batch size is large, the optimal learning rates of SGD with Momentum and warmup learning rates are higher than the optimal learning rate for SGD, and we see Momentum and warmup learning rates start outperforming vanilla SGD. These results are entirely consistent with the two regimes of SGD and SGD with Momentum discussed in section 2.

**16-4 Wide-ResNet on CIFAR without batch normalization.** In figure 5 we present results when training our 16-4 Wide-ResNet (Zagoruyko & Komodakis, 2016). We follow the same setup and learning rate schedule described in section 3, and we train for 200 epochs. However we remove batch normalization, and introduce ZeroInit (described in appendix C). The performance of SGD degrades with increasing batch size on both the test set and the training set, while the performance of Momentum with or without learning rate warmup is constant for batch sizes $B \lesssim 256$. Above this threshold, the performance of both methods degrades rapidly. These observations are explained by the optimal effective learning rates in figure 5c. Momentum has similar optimal learning rates with and without warmup. In both cases the learning rate initially increases proportional to the batch size before saturating. The optimal learning rate of SGD is curvature bound at all batch sizes considered.

**ResNet-50 on ImageNet.** In table 3, we provide results for ResNet-50 trained on ImageNet for 90 epochs at a small range of batch sizes. We follow the modified ResNet-50 implementation of Goyal et al. (2017), and we use our standardized learning rate schedule without warmup (see appendix A). We train a single model at each batch size-learning rate pair. SGD with and without Momentum achieve similar test accuracies when the batch size is small, but SGD with Momentum outperforms SGD without Momentum when the batch size is large. The optimal effective learning

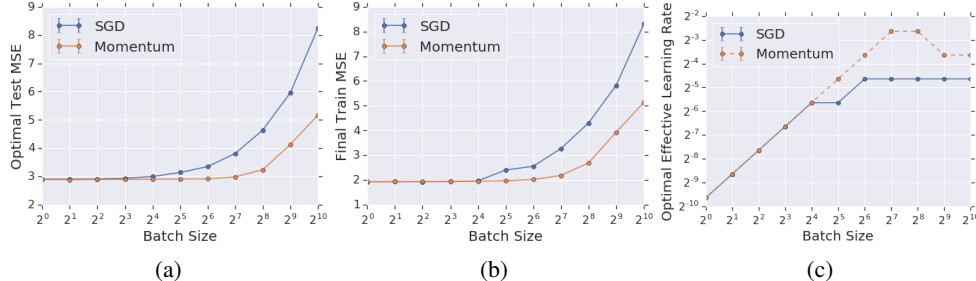

(a)           (b)           (c)

Figure 6: A fully connected autoencoder, trained on MNIST for 200 epochs. We report the performance of SGD and SGD w/ Momentum. We perform a grid search to identify the optimal learning rate which maximizes the mean-squared error (MSE) on the test set, and report the mean performance of the best 5 of 7 runs. a) The test MSE of SGD w/ Momentum is initially independent of batch size, but it begins to rise when the batch size exceeds 128. The test MSE of vanilla SGD starts rising for batch sizes exceeding 16. b) We see similar phenomena on the training set MSE. c) The optimal effective learning rate is proportional to batch size when the batch size is small for both vanilla SGD and SGD w/ Momentum, while it becomes independent of batch size for larger batch sizes. The optimal effective learning rate in the curvature dominated regime is larger for SGD w/ Momentum.

rate is proportional to batch size for all batch sizes considered when using SGD with Momentum, but not when using SGD without Momentum.

**Fully connected autoencoder on MNIST.** In figure 6, we present results when training a fully-connected autoencoder on the MNIST dataset (Sutskever et al., 2013). Our network architecture is described by the sequence of layer widths $\{784, 1000, 500, 250, 30, 250, 500, 1000, 784\}$, where 784 denotes the input and output dimensions. For more details on this architecture, refer to Sutskever et al. (2013). Because of the bottleneck structure of the model, it is known to be a difficult problem to optimize and has often been used as an optimization benchmark (Sutskever et al., 2013; Kidambi et al., 2018). The L2 regularization parameter was set at $10^{-5}$. We use the learning rate decay schedule described in appendix A without learning rate warmup, and we train each model for 200 epochs. As before, we notice that for small batch sizes, the performance of both SGD and SGD w/ momentum is independent of batch size, while performance begins to degrade when the batch size is large. On this model, the performance of SGD begins to degrade at much smaller batch sizes than we observed in residual networks, and consequently SGD w/ momentum starts outperforming SGD at much smaller batch sizes. This is likely due to the poor conditioning of the model due to the bottleneck structure of its architecture.

**LSTM on Penn TreeBank.** Finally in figure 7, we present results when training a word-level LSTM language model on the Penn TreeBank dataset (PTB), following the implementation described in

Table 3: ResNet-50, trained on ImageNet for 90 epochs. We follow the implementation of Goyal et al. (2017), however we introduce our modified learning rate schedule defined in appendix A. We do not use learning rate warmup. We perform a grid search to identify the optimal effective learning rate and report the performance of a single training run. The test accuracies achieved by SGD and Momentum are equal when the batch size is small, but Momentum outperforms SGD when the batch size is large. For SGD with Momentum, the optimal effective learning rate is proportional to batch size for all batch sizes considered, while this linear scaling rule breaks at large batch sizes for SGD.

|  | Batch size | Optimal test accuracy (%) | Training loss | Optimal effective learning rate |
|---|---|---|---|---|
| SGD | 256 | 77.0 | 2.25 | 1.0 |
| | 1024 | 76.7 | 2.25 | 4.0 |
| | 4096 | 76.1 | 2.30 | 8.0 |
| Momentum | 256 | 77.0 | 2.25 | 1.0 |
| | 1024 | 76.8 | 2.25 | 4.0 |
| | 4096 | 76.8 | 2.25 | 16.0 |

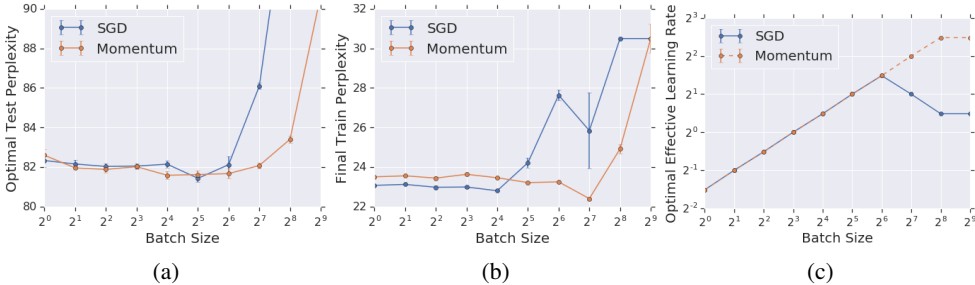

(a)  (b)  (c)

Figure 7: A word-level LSTM language model trained on PTB for 40 epochs. We report the performance of SGD and SGD w/ Momentum. We perform a grid search to identify the optimal learning rate which maximizes the test set perplexity, and report the mean performance of the best 5 of 7 runs. a) The test set perplexity of SGD w/ Momentum is independent of batch size when the batch size is small, but begins to rise when the batch size exceeds 128. The test set perplexity of vanilla SGD starts rising for batch sizes exceeding 64. b) We see similar phenomena on the training set perplexity. c) The optimal effective learning rate is proportional to *square root* of the batch size when the batch size is small, while it levels off for larger batch sizes. The gradients of consecutive minibatches in a language model are not independent, violating the assumptions behind linear scaling.

Zaremba et al. (2014). The LSTM used has two layers with 650 units per layer. The parameters are initialized uniformly in $[-0.05, 0.05]$. We apply gradient clipping at 5, as well as dropout with probability 0.5 on the non-recurrent connections. We train the LSTM for 40 epochs using an unroll step of 35, and use the learning rate decay schedule described in appendix A without learning rate warmup. As with the other models tested in this paper, this learning rate schedule reaches the same test perplexity performance as the original schedules reported in Zaremba et al. (2014). Once again, we see that SGD and SGD w/ momentum have similar performance for small batch sizes. Performance for SGD starts degrading for batch sizes exceeding 64, whereas performance for SGD w/ momentum starts degrading for batch sizes exceeding 128. However, as mentioned in section 3, the optimal learning rate increases as *square root* of the batch size for small batch sizes, before leveling off at a constant value for larger batch sizes. This is likely due to correlations in consecutive data samples when training the LSTM, which violate the assumptions used to derive the linear scaling rule in section 2.

# E    ADDITIONAL RESULTS UNDER A CONSTANT STEP BUDGET

Here we provide some additional results studying how the optimal test accuracy depends on the batch size under a constant step budget. In table 4, we train a fully connected auto-encoder on MNIST for 156,250 updates (Sutskever et al., 2013). This corresponds to 200 epochs when the batch size is 64. We described this model in appendix D, and we train using the learning rate schedule defined in appendix A using SGD with Momentum without learning rate warmup. The test set MSE increases as the batch size increases, while the training set MSE falls as the batch size rises. Although the training set MSE appears to rise for a batch size of 4096, this only occurs because the optimal effective learning rate is measured on the test set. The optimal effective learning rate is independent of the batch size, suggesting that the learning rate may be close to curvature dominated regime.

In table 5, we train a word-level LSTM on the Penn TreeBank (PTB) dataset (Zaremba et al., 2014) for 16560 updates. This corresponds to 40 epochs when the batch size is 64. We described this model in appendix D, and we train using the learning rate schedule defined in appendix A using SGD with Momentum without learning rate warmup. The test perplexity increases significantly as the batch size increases, while the training perplexity falls as the batch size rises. The optimal effective learning rate increases as the batch size rises, suggesting that we are inside the noise dominated regime.

Table 4: The optimal test set MSE and final training set MSE for a range of batch sizes under a constant step budget. For each batch size, we train a fully connected autoencoder on MNIST for 156,250 updates. We perform a grid search to identify the optimal learning rate which maximizes the test set MSE, and we provide the average performance of the best 5 out of 7 training runs. The final test MSE falls for large batch sizes, although this effect is rather weak in this model.

| Batch size | Optimal test set MSE | Final training set MSE | Optimal effective learning rate |
|---|---|---|---|
| 64 | $2.91 \pm 0.01$ | $2.017 \pm 0.003$ | 0.08  (0.08 to 0.08) |
| 256 | $2.95 \pm 0.01$ | $2.010 \pm 0.005$ | 0.08  (0.08 to 0.08) |
| 1024 | $2.96 \pm 0.01$ | $2.005 \pm 0.011$ | 0.08  (0.08 to 0.08) |
| 4096 | $2.98 \pm 0.01$ | $2.018 \pm 0.008$ | 0.08  (0.08 to 0.08) |

Table 5: The optimal test set perplexity and final training set perplexity for a range of batch sizes under a constant step budget. For each batch size, we train a word-level LSTM on PTB for 16560 updates. We perform a grid search to identify the optimal learning rate which maximizes the test set perplexity, and we provide the average performance of the best 5 out of 7 training runs. The optimal test perplexity increases considerably as the batch size rises, while the training perplexity falls.

| Batch size | Optimal test perplexity | Final training perplexity | Optimal effective learning rate |
|---|---|---|---|
| 64 | $81.67 \pm 0.26$ | $23.25 \pm 0.05$ | 2.8  (2.8 to 2.8) |
| 128 | $86.04 \pm 0.49$ | $21.16 \pm 0.06$ | 5.6  (5.6 to 5.6) |
| 256 | $92.19 \pm 0.26$ | $15.74 \pm 0.03$ | 5.6  (5.6 to 5.6) |

# F  ADDITIONAL RESULTS WITH A FIXED BATCH SIZE AND VARIABLE EPOCH BUDGET

We now provide additional experimental results to accompany those provided in section 5, where we study whether the optimal training temperature is independent of the epoch budget. We use SGD with Momentum with the momentum parameter $m = 0.9$ for all our experiments in this section. In figure 8, we present results on a word-level LSTM on the PTB dataset for a batch size of 64 and for varying epoch budgets. Note that the original LSTM model in Zaremba et al. (2014) was trained for 39 epochs. The results in figure 8 are remarkably similar to those presented in figure 2. As the epoch budget rises, the test set perplexity first falls but then begins to increase. The training set perplexity falls monotonically as the epoch budget increases. Finally, the optimal learning rate which minimizes the test set perplexity is independent of the epoch budget once this epoch budget is not too small, while the optimal learning rate which minimizes the training set perplexity falls.

In figure 9, we present results on a fully connected autoencoder trained on MNIST for a batch size of 32 and for a range of epoch budgets. Note that the autoencoder results presented in section D were trained for 200 epochs. Figures 9a and 9b are similar to figures 2a and 2b in the main text. Initially the test set MSE falls as the epoch budget increases, but then it starts increasing. The training set MSE falls monotonically as the epoch budget rises. In figure 9c however, we notice that the learning rate that minimizes the test set MSE decreases as the epoch budget rises. This is the opposite of what we observed in figures 2 and 8. To further investigate this, in figure 9d we plot the mean test set MSE during training for an epoch budget of 800 for learning rates $\epsilon = 0.004$ and $\epsilon = 0.002$. We notice that for the larger learning rate $\epsilon = 0.004$, the model overfits faster on the training set, causing the test set MSE to rise by the time of the first learning rate drop at 400 epochs. This is consistently the case for all epoch budgets over 200 epochs. To avoid the test set MSE from rising, the optimal learning rate for the test MSE drops to slow down training sufficiently such that there is no overfitting before the first learning rate decay. Meanwhile the optimal learning rate to minimize the training loss is more or less constant. This suggests that early stopping is particularly important in this architecture and dataset, and that it has more influence on the final test performance than stochastic gradient noise.

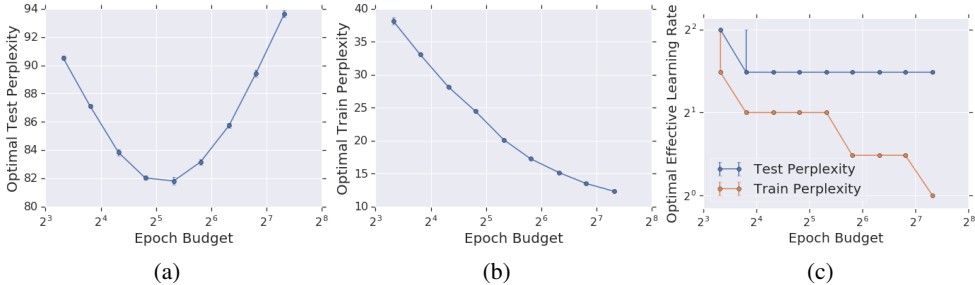

(a)          (b)          (c)

Figure 8: The performance of a word-level LSTM language model trained on the Penn TreeBank dataset using SGD with Momentum and a batch size of 64 at a range of epoch budgets. We identify both the optimal effective learning rate which minimizes the test set perplexity and the optimal effective learning rate which minimizes the training set perplexity, and we present the mean performance of the best 5 out of 7 runs. a) Initially the test set perplexity falls as the epoch budget increases, however it begins to rise beyond 56 training epochs. b) The training set perplexity falls monotonically as the epoch budget rises. c) The learning rate that minimizes the training set perplexity falls as the epoch budget rises, while the learning rate that minimizes the test set perplexity only varies by a factor of 2 when the epoch budget rises over two orders of magnitude.

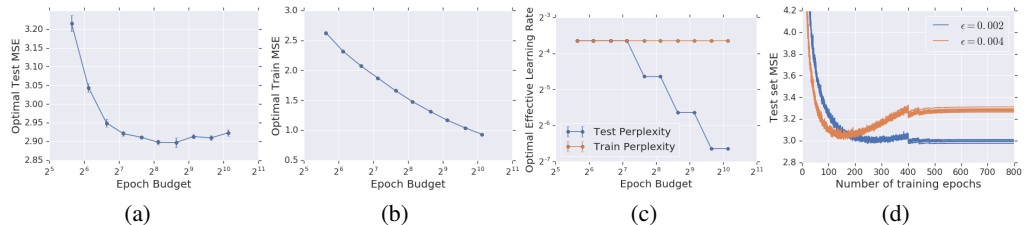

(a)      (b)      (c)      (d)

Figure 9: The performance of a fully connected autoencoder on MNIST using SGD with Momentum and a batch size of 32 with varying training epoch budget. We identify both the optimal effective learning rate which minimizes the test set MSE and the optimal effective learning rate which minimizes the training set MSE, and we present the mean performance of the best 5 out of 7 runs. a) Initially the test set MSE falls as the epoch budget increases, and it only starts going up very slightly for large epoch budgets. b) The train set MSE falls monotonically as the the epoch budget rises. c) The learning rate that minimizes the test set MSE decreases, while the learning rate that minimizes the train set MSE remains constant as the epoch budget rises. This is contrary to what we observe in figures 2 and 8. The reason for this is apparent from figure d), where we plot the test set MSE during training for all 7 runs for an epoch budget of 800 for learning rate $\epsilon = 0.004$ and $\epsilon = 0.002$. We notice that for a larger learning rate, the model overfits on the training set faster, causing the test set MSE to rise by the time of the first learning rate drop at 400 epochs. This suggests that early stopping has more influence on the final test performance in this architecture than stochastic gradient noise.

## G   ADDITIONAL RESULTS SWEEPING BOTH THE INITIAL AND FINAL LEARNING RATES

In section 5 and appendix F, we study how the optimal learning rate depends on the epoch budget. In these experiments, we find evidence that there may be an optimal temperature early in training which is beneficial for good generalization performance on classification tasks. However the learning rate schedules used for these experiments have the property that the initial learning rate, which we denote in this section with $\epsilon_0$, is coupled with the final learning rate, which we denote in this section with $\epsilon_f$. More specifically, the final learning rate $\epsilon_f = \epsilon_0 \cdot \gamma^{-10}$, where $\gamma$ denotes the decay factor, which we set to 2 in the bulk of our experiments (this schedule is described in detail in section A).

Coupling the initial and final learning rates make it less clear whether this optimal temperature is important at the start or the end of training, and it is also not clear to what extent our conclusions

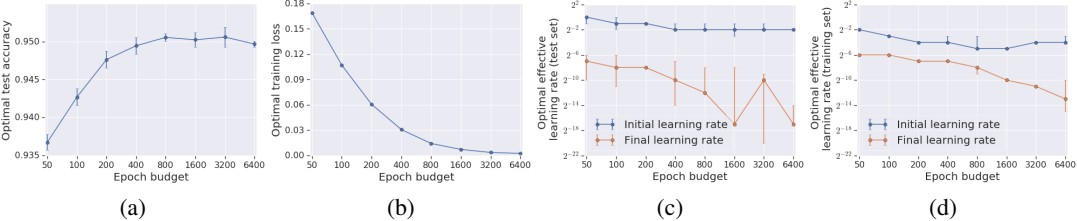

(a)  (b)  (c)  (d)

Figure 10: Performance of a 16-4 Wide-ResNet with batch normalization trained on CIFAR-10 using SGD with Momentum at a batch size of 64. We tune the initial and the final learning rates independently, as described in section G. We plot both the optimal initial and final learning rates for maximizing the test set accuracy, as well as the optimal initial and final learning rates for minimizing the training set loss, and we present the mean performance of the best 5 out of 7 runs. a) The test accuracy initially increases with increasing compute budget before saturating for epochs budgets greater than 800. b) Meanwhile the training loss falls monotonically as the epoch budget rises. c) The optimal initial learning rate which maximizes the test accuracy is constant for epoch budgets greater than 400, while the optimal final learning rate decays rapidly as the epoch budget increases. d) The optimal initial learning rate which minimizes the training loss decays slowly as the epoch budget increases, while the optimal final learning rate decays more rapidly.

are influenced by our choice of decay factor. Therefore in this section, we perform experiments on varying epoch budgets where we tune both the initial and the final learning rates independently. As in the schedule presented in section A, when training for an epoch budget of $N_{epochs}$, we use the initial learning rate for the first $N_{epochs}/2$ epochs, and we then decay the learning rate by a factor of $\gamma$ every $N_{epochs}/20$. To define $\gamma$, we select both an initial learning rate $\epsilon_0$ and a final learning rate $\epsilon_f$, and we then set $\gamma = (\epsilon_0/\epsilon_f)^{1/10}$. We do not use learning rate warmup. These experiments require a very large compute budget, and so we only study our 16-4 Wide-ResNet model with batch normalization.

In figure 10, we show results when training this 16-4 Wide-Resnet on CIFAR-10 at a batch size of 64 using SGD with Momentum. We train for a range of epoch budgets from 50 to 6400 epochs, and we evaluate the optimal initial and final learning rates independently for both maximizing the test set accuracy and minimizing the training loss. From figure 10a and 10b, we see the same trends as observed in section 5 and appendix F. Specifically, when we increase the compute budget, the optimal test set accuracy first increases, and then saturates for epoch budgets greater than 800, while the optimal training loss falls monotonically as the epoch budget grows.

In figures 10c and 10d, we plot the optimal initial and final learning rates for maximizing the test set accuracy (10c) and minimizing the training set loss (10d). We make several observations from these plots. First, the optimal initial learning rate for maximizing the test set accuracy decays very slowly as the epoch budget rises, and it is constant for epoch budgets greater than 400. This supports the existence of an optimal temperature early in training which boost generalization performance. Meanwhile, the optimal final learning rate for maximizing the test set accuracy does decay rapidly as the epoch budget increases, which is likely helpful to prevent overfitting at late times. We note that the error bars on the final learning rate are much larger than those on the initial learning rate, suggesting that it is the initial learning rate which is most important to tune in practice.

The optimal initial learning rate for minimizing the training loss also decays slowly as the epoch budget rises (decreasing by a factor of 4 to 8 when the epoch budget rises by a factor of 128), while the optimal final learning rate for minimizing the training loss decays much more quickly (roughly by a factor of 128 over the same range). The optimal initial learning rate for maximizing the test accuracy is consistently higher than the optimal initial learning rate for minimizing the training loss, while the optimal final learning rate for maximizing the test accuracy is consistently lower than the optimal final learning rate for minimizing the training loss. These two observations support the widely held belief that learning rate schedules which maintain a high temperature at early times, and then decay the learning rate rapidly at late times generalize well to the test set in some architectures.

