# OpenReview forum: "Hyperparameter Tuning and Implicit Regularization in Minibatch SGD"
_ICLR.cc/2020/Conference — Reject_

### Official Review · AnonReviewer1 · 2019-10-18
**Official Blind Review #1**

**Rating:** 3

**Review:**

The paper attempts to clarify the debate on large-batch neural network training, particularly on the relationship between learning rate, batch sizes and test performance. The authors claim two contributions towards understanding how the hyper-parameters of SGD affect final training and test performance: (1) SGD exhibits two regimes with different behaviours and (2) large-batch training leads to degradation of test performance even with same step budgets.

Overall, the authors did a comprehensive study on large-batch training with the support of extensive experiments. But I'm concerned with the novelty and contributions of this paper. I tend to reject this paper because (1) the first contribution of the paper is not new as it has already been recognized by a few paper that SGD exhibits two different regimes; (2) this paper makes the debate of large-batch training even muddier.

Main argument:
The paper does not do a great job in clarify the debate. Particularly, the authors mixed their observations up with the results of published works, making it hard to identify the contributions of this paper. For example, the two regimes mentioned in the paper has been identified by a few other works and the contribution of this paper is just to verify them again. Also, I find the experiments done in section 3 and 4 are similar to previous works and even the conclusions are similar. The only new observation I'm aware of in these two sections is that the training loss and test accuracy are independent of batch size in the noise dominated regime.

Back to introduction section, the goal of this paper (as claimed in the beginning of second paragraph) is to clarify the debate. But does this paper really achieves this goal? In terms of learning rate scaling, this paper gets similar conclusions as Shallue et al. (2018). In terms of the difference between vanilla SGD and SGD with momentum, Zhang et al. (2019) already argued that the difference depends on specific batch sizes and SGD with momentum only outperforms SGD in the curvature dominated regime.

I think the authors should instead focus on the discussion of generalization performance and the observation that training loss and test accuracy are independent of batch size in noise dominated regime. To my knowledge, this part is novel and interesting.

In summary, I'm inclined to reject this paper given the current version. However, I think the paper is still worth reading if the authors can reorganize the paper and I might increase my score if my concerns get resolved.


**Experience Assessment:**

I have published one or two papers in this area.

**Review Assessment: Checking Correctness Of Derivations And Theory:**

I assessed the sensibility of the derivations and theory.

**Review Assessment: Checking Correctness Of Experiments:**

I assessed the sensibility of the experiments.

**Review Assessment: Thoroughness In Paper Reading:**

I read the paper at least twice and used my best judgement in assessing the paper.

---

> ### Author Response · Authors · 2019-11-08
> **Response to review**
>
> We thank the reviewer for their helpful comments.
>
> Please could the reviewer clarify why they felt our work muddies the debate regarding large-batch training? We demonstrate that one can initially increase the batch size with no loss in test accuracy by simultaneously increasing the learning rate. However for very large batch sizes the test accuracy degrades under both constant epoch and constant step budgets.
>
> We agree that some of our observations under constant epoch budgets in sections 2 and 3 have been made in previous work. However there are also several important differences:
>
> 1. Our paper is the first to relate the two regimes of SGD to the popular analogy between SGD and stochastic differential equations (SDEs). As we show in sections 4 and 5, this perspective is crucial to understanding the influence of batch size and learning rate on test accuracy. A common criticism of this analogy is that SGD noise is not Gaussian when the batch size is small. To our knowledge, we are the first to show that the analogy between SGD and SDEs holds for non-Gaussian short-tailed noise (appendix B).
>
> 2. Zhang et al. argued that Momentum only helps in the large batch limit. However, their analysis is based on the noisy quadratic model, which cannot explain the results we observed on the test set in sections 4 and 5. These experiments clearly demonstrate that, unlike the SDE perspective, the noisy quadratic model is not an appropriate model for predicting test set performance in deep learning. Their work also does not clarify the assumptions under which linear scaling of the learning rate should arise.
>
> 3. Our empirical results in section 3 are similar to Shallue et al., however their work argues that there is no reliable relationship between learning rate and batch size. We draw a very different conclusion: the learning rate usually obeys linear scaling, but linear scaling only holds theoretically when the assumptions we specify are satisfied. Linear scaling may not hold in cases where these assumptions break down (e.g., language modelling).
>
> 4. The observation that the test accuracy is independent of batch size in the noise dominated regime is a natural consequence of the SDE analogy, since any two training runs which integrate the same SDE should sample final parameters from the same probability distribution. We will clarify this in the updated text.
>
> Two reviewers complained that it was difficult to tell from the text which contributions are novel and which also appear in previous works. We apologise for this. It was not our intention and we will edit sections 1 and 2 to ensure that this is resolved and that the above points are reflected in the text.
>
> Turning to our generalization experiments in sections 4 and 5. It is true that a number of papers in recent years have claimed that SGD noise enhances generalization. However Shallue et al. recently argued no previous work had provided convincing empirical evidence for this claim. Indeed in their abstract, they state ‘We find no evidence that larger batch sizes degrade out-of-sample performance’. In another recent paper, Zhang et al. argued that optimization in deep learning is well described by a noisy quadratic model which predicts that increasing the batch size should always enhance performance under constant step budgets.
>
> Crucially, to establish that SGD noise enhances generalization, one must show that small batch sizes generalize better than large batch sizes under constant step budgets, with realistic learning rate decay schedules, and one must independently tune the learning rate at each batch size. In section 4, we are the first authors to perform this experiment and confirm that the final test accuracy of SGD does degrade for very large batch sizes under both constant epoch and constant step budgets, contradicting the claims of both Shallue et al and Zhang et al. Furthermore, we show in section 5 that the optimal SGD temperature which maximizes the test accuracy is almost independent of the epoch budget. These results provide the first convincing empirical evidence that SGD noise does enhance generalization in well-tuned networks with learning rate decay schedules. We believe this is an important contribution.

---

> > ### Comment · AnonReviewer1 · 2019-11-13
> > **Thank you for the response.**
> >
> > I've read your response and my score remains unchanged because I haven't seen any update of the paper.

---

### Official Review · AnonReviewer2 · 2019-10-23
**Official Blind Review #2**

**Rating:** 3

**Review:**

This paper studies the properties of SGD as a function of batch size and learning rate. Authors argue that SGD has two regimes:  a noise dominated regime (small batch size) and curvature dominated regime (large batch size). Authors conduct through numerical experiments highlighting how learning rate changes as a function of batch size (initially linear growth and then saturates). The critical contribution of this work appears to be the observation that large batch size can be worse than small under same number of steps demonstrating implicit regularization of small batch size.

The two regime claim of the paper is not really novel. These regimes are fairly well covered by previous works (e.g. Belkin et al as well as others). When it comes to experiments, constant epoch budget is also fairly well understood and the behavior in Figure 1 is not really surprising (as the eventual training performance gets worse with large batches).

The interesting part in my opinion is the experiments on constant steps. Authors verify large batch size reduces test accuracy while improving train. I believe these experiments are novel and the results are interesting. Besides CIFAR 10, authors test this hypothesis in two other datasets while tuning the learning rate. On the other hand, contribution is somewhat incremental given observations made by related literature (Keskar et al and others).

Some remarks:
1) In Table 1, batch size 16k has effective LR of 32. However in Figure 1c SGD with momentum at batch size 8k uses an effective LR of 4. Can you explain this inconsistency i.e. why is there such a huge jump from 4 to 32 (in reality we expect the effective LR to stay constant in the curvature regime). I also understand that one is constant epoch and other is constant step. However 4 to 32 seems a bit inconsistent.

2) Does momentum help in constant step budget (with sufficiently large steps so that training loss is small)?

3) Readability: Consider explaining what is meant by "warm-up", "epoch budget", "step budget" clearly and upfront.


**Experience Assessment:**

I have read many papers in this area.

**Review Assessment: Checking Correctness Of Derivations And Theory:**

N/A

**Review Assessment: Checking Correctness Of Experiments:**

I assessed the sensibility of the experiments.

**Review Assessment: Thoroughness In Paper Reading:**

I read the paper at least twice and used my best judgement in assessing the paper.

---

> ### Author Response · Authors · 2019-11-08
> **Response to review**
>
> We thank the reviewer for their helpful comments.
>
> We agree that our most surprising results are for SGD under constant step budgets or unlimited epoch budgets. However the behaviour of SGD under constant epoch budgets has generated a lot of debate in the literature in recent years, and we felt it was important to address this simple case first. We agree that some of the observations in sections 2 and 3 have already been made in previous work, however there are also several important differences:
>
> 1. Ma, Bassily and Belkin also introduced the notion of two regimes, however their theory holds for convex losses in the interpolating regime. We will discuss their contribution explicitly in the updated text. Our discussion in section 2 clarifies why the two regimes arise in practical deep learning models for which these conditions may not hold.
>
> 2. Our paper is the first to relate the two regimes of SGD to the popular analogy between SGD and stochastic differential equations (SDEs). As we show in later sections, this perspective is crucial to understanding the influence of batch size and learning rate on test accuracy. A common criticism of this analogy is that SGD noise is not Gaussian when the batch size is small. To our knowledge, we are the first to show that the analogy between SGD and SDEs holds for non-Gaussian short-tailed noise (appendix B).
>
> 3. We clarify the differences to some other recent papers in our reply to reviewer 1.
>
> Two reviewers complained that it was difficult to tell from the text which contributions are novel and which also appear in previous works. We apologise for this. It was not our intention and we will edit sections 1 and 2 to ensure that this is resolved and that the above points are reflected in the text.
>
> Turning to our generalization experiments in sections 4 and 5. We agree that many authors have proposed that SGD noise enhances generalization. Most notably, Keskar et al. argued that large minibatches perform worse than small minibatches on the test set, even when both achieve similar performance on the training set. However their experiments do not provide convincing evidence for this claim, because they tuned the learning rate with small batches and then used the same learning rate value with large batches. A convincing experiment should independently tune the learning rate at all batch sizes under a constant step budget, and it should use a realistic learning rate decay schedule.
>
> Indeed, Shallue et al. recently argued that no existing paper has provided convincing evidence that small batch sizes generalize better than large batch sizes under constant step budgets, and they state in their abstract ‘We find no evidence that larger batch sizes degrade out-of-sample performance’. Meanwhile, Zhang et al. argued that optimization in deep learning is well described by a noisy quadratic model which predicts that increasing the batch size should always enhance performance under constant step budgets. To our knowledge, our experimental results in section 4 are the first to provide convincing evidence that very large minibatches do perform worse than small batch sizes on the test set, even under constant step budgets and when the learning rate is independently tuned. We believe this is an important contribution. Meanwhile, our results in section 5 suggest that SGD has an optimal temperature early in training which promotes generalization and is independent of the epoch budget.
>
> In response to the reviewer’s specific comments:
>
> 1) Looking at Figure 1c, while the optimal learning rate at 8k with Momentum is 4, the error bars at this batch size range from 4 to 32. These error bars can be very large in the curvature regime, precisely because the optimal learning rate is close to instability.
>
> 2) Yes, Momentum will help under constant step budgets if the batch size is large, since it enables us to achieve larger effective learning rates which are beneficial for generalization. We will add additional experiments to the text to clarify this.
>
> 3) We will clarify the meaning of warm up, epoch budget and step budget as requested.

---

### Official Review · AnonReviewer3 · 2019-11-04
**Official Blind Review #3**

**Rating:** 3

**Review:**

This paper is an empirical contribution regarding SGD arguing that it presents two different behaviors which the authors name a noise dominated regimen, and a curvature dominated regime. They observe that the behaviors seem to arise in different batch sizes

The authors derive empirical conclusions and perform experiments in different settings. The paper is well-written and the experimental setup seems to be carefully carried out.

I find the observations interesting, but the contribution is empirical and not entirely new. It would be nice if there were some theoretical results to back up the observations.

**Experience Assessment:**

I have read many papers in this area.

**Review Assessment: Checking Correctness Of Derivations And Theory:**

N/A

**Review Assessment: Checking Correctness Of Experiments:**

I assessed the sensibility of the experiments.

**Review Assessment: Thoroughness In Paper Reading:**

I read the paper at least twice and used my best judgement in assessing the paper.

---

> ### Author Response · Authors · 2019-11-08
> **Response to review**
>
> We thank the reviewer for their comments.
>
> Although our primary contributions are empirical, we also provided a detailed theoretical discussion in section 2, where we give a clear and simple account of why the two regimes arise. Although previous authors have also discussed some of these results, there are differences between our conclusions, as we discussed in our responses to the other two reviewers.
>
> We would also like to emphasize that we make a significant contribution to the debate regarding SGD and generalization. While many papers have proposed that small batches may generalize better than large minibatches, it was recently pointed out by Shallue et al. that none of these experiments provide convincing evidence for this claim, because no experiment to date has compared small and large batch training under a constant step budget with a realistic learning rate decay schedule while independently tuning the learning rate at each batch size. We are the first to run this experiment and conclusively establish that SGD noise does enhance generalization in popular models/datasets. We believe this is an important contribution.
>
> We also provide intriguing results as we vary the epoch budget, which demonstrate that the optimal learning rate which maximizes the test accuracy does not decrease as the epoch budget rises. This supports the notion that SGD has an optimal “temperature” which biases it towards solutions that generalize well. Additional experiments in the appendix G go further and study how the optimal learning rate schedule changes as we increase the epoch budget.

---

### Public Comment · ~Guodong_Zhang1 · 2019-10-03
**Minor Comments**

Hi,

In terms of two points (in the second paragraph) you made in the intro, I think you need to cite previous work properly.

1. "In the noise dominated regime, the final training loss and test accuracy are independent of
batch size under a constant epoch budget, and the optimal learning rate increases as the
batch size rises. In the curvature dominated regime, the optimal learning rate is independent
of batch size, and the training loss and test accuracy degrade with increasing batch size. The
critical learning rate which separates the two regimes varies between architectures."

You should give credits to previous work on that as it's not a new observation.

3. "SGD with Momentum and learning rate warmup do not outperform vanilla SGD in the noise
dominated regime, but they can outperform vanilla SGD in the curvature dominated regime."

My paper "Which Algorithmic Choices Matter at Which Batch Sizes? Insights From a Noisy Quadratic Model" already made this point empirically and theoretically with some assumptions.

---

> ### Author Response · Authors · 2019-10-14
> **Response to comments**
>
> There are many theoretical and empirical papers on this topic, however we believe there is not yet consensus in the community. As we emphasized in the introduction of our paper, our main contribution is to provide clarity with substantial empirical evidence supporting both the existence of two distinct SGD regimes, as well as the existence of implicit regularization arising from the noise in the gradient estimate. As we mention in the paper, some of the theoretical predictions we discuss have been known for a long time and derived under multiple different assumptions, and we have tried to cite multiple papers for each claim where appropriate.
>
> We cite a number of papers which discuss the notion that SGD exhibits qualitatively different behaviors at different batch sizes in the paragraph immediately preceding the bullet points that you mention. We are happy to add your paper to this list. Most recent theory papers in deep learning have focused on the behavior of SGD in the small batch "noise dominated" regime, and we cite these appropriately when we discuss this regime in depth in section 2.
>
> It is well known that full batch Momentum converges faster than gradient descent, and there are a number of papers from the 90s onward which prove that SGD and Momentum are equivalent in the small batch small learning rate limit so long as the momentum coefficient is not too large. We cite many of these papers in section 3, and we also mention that Shallue et al. observed this phenomenon empirically last year. We are happy to include your paper in this list too.
>
> We would like to clarify that we do already cite your work multiple times in the main text, including in the introduction when we state that many of our theoretical results can be derived from different assumptions.

---

### Decision · Program_Chairs · 2019-12-19

**Decision:**

Reject

**Comment:**

Authors provide an empirical evaluation of batch size and learning rate selection and its effect on training and generalization performance. As the authors and reviewers note, this is an active area of research with many closely related results to the contributions of this paper already existing in the literature. In light of this work, reviewers felt that this paper did not clearly place itself in the appropriate context to make its contributions clear. Following the rebuttal, reviewers minds remained unchanged.